# Value of Left Atrial Appendage Function Measured by Transesophageal Echocardiography for Prediction of Atrial Fibrillation Recurrence after Radiofrequency Catheter Ablation

**DOI:** 10.3390/diagnostics11081465

**Published:** 2021-08-13

**Authors:** Sabina Istratoaie, Ștefan C. Vesa, Gabriel Cismaru, Dana Pop, Radu Roșu, Mihai Puiu, Diana Pepine, Cristina Ciobanu, Ioan A. Minciuna, Gelu Simu, Dumitru Zdrenghea, Anca D. Buzoianu

**Affiliations:** 1Department of Pharmacology, Toxicology and Clinical Pharmacology, Iuliu Haţieganu University of Medicine and Pharmacy, 400337 Cluj-Napoca, Romania; sabina.istratoaie@gmail.com (S.I.); stefanvesa@gmail.com (Ș.C.V.); abuzoianu@umfcluj.ro (A.D.B.); 25th Department of Internal Medicine, Cardiology-Rehabilitation, Iuliu Haţieganu University of Medicine and Pharmacy, 400066 Cluj-Napoca, Romania; gabi_cismaru@yahoo.com (G.C.); rosu.radu1053@gmail.com (R.R.); puiu.mihai@yahoo.com (M.P.); diana.pepine@yahoo.com (D.P.); cristina.cbn@yahoo.com (C.C.); iaminciuna@gmail.com (I.A.M.); simugelu@yahoo.com (G.S.); dzdrenghea@yahoo.com (D.Z.)

**Keywords:** atrial fibrillation, transesophageal echocardiography, left atrial appendage emptying flow velocity

## Abstract

Atrial fibrillation (AF) recurrence after radiofrequency catheter ablation (RFCA) remains a challenging issue. This study aims to explore the left atrial appendage function by transesophageal echocardiography (TEE) and assess its value in predicting AF recurrence following RFCA in paroxysmal AF patients. Eighty-one patients with paroxysmal AF that underwent RFCA were recruited. TEE was performed before ablation with the assessment of left atrial appendage emptying flow velocity (LAAeV). AF recurrence occurred in 24 patients (29.6%) within 12 months after RFCA. The left atrium diameter (LAD) and left atrium volume index (LAVI) were both significantly higher in the recurrence group compared to the non-recurrence group, while the LAAeV was significantly lower in the recurrence group. LAD, LAVi and LAAeV were univariately significant risk factors for AF recurrence after ablation. Based on receiver operating curve (ROC), LAAeV < 40.5 cm/s, LAVi > 40.5 mL and LAD > 41 mm were identified as cut-off values for predicting AF recurrence. In multivariate regression analysis LAAeV < 40.5 cm/s (HR 8.194, 95% CI 2.980–22.530, *p* < 0.001) was identified as the only statistically significant independent predictor of AF recurrence, as the statistical significance threshold was not achieved for LAVI > 40.5 mL and LAD > 41 mm (*p* = 0.319; *p* = 0.507, respectively). A low LAAeV was the only important independent predictor of AF recurrence within 1 year after first RFCA.

## 1. Introduction

Atrial fibrillation (AF) is the most common sustained cardiac arrhythmia, with a lifetime AF risk of about one in three individuals of European ancestry over 55 years of age [1,2]. The presence of AF is associated with impaired quality of life and increased risk of stroke, heart failure and cardiovascular mortality. RFCA targeting pulmonary vein isolation is a well established effective therapeutic option for paroxysmal AF. However, the success rate of PVI ranges between 50 and 80% [3,4]. Therefore, evaluation of predictive factors of AF recurrence is important to optimize the selection of appropriate patients as well as to improve the success rate of RFCA. It has been reported that left atrial dilation and decreased left atrium (LA) function is correlated with a high AF recurrence rate [5,6,7]. In addition to left atrial appendage (LAA) playing important role in predicting cardioembolic stroke, LAA can also reflect LA function and severity of the LA remodeling in patients with AF [8,9]. However, information on parameters to assess the role of LAA function in AF recurrence after RFCA is limited. The imaging technique of first choice to evaluate LAA structure and function is transesophageal echocardiography. The function of the LAA is most commonly determined by measuring emptying velocity with pulsed-wave Doppler. The purpose of this study was to assess whether left atrial appendage emptying flow velocity (LAAeV) may provide predictive value for paroxysmal AF recurrence within 1 year after first RFCA.

## 2. Materials and Methods

### 2.1. Patient Selection

Patients with symptomatic, paroxysmal AF who had undergone the first RFCA for AF in the Cardiology Department of the Rehabilitation Hospital from Cluj-Napoca, between December 2018 and January 2020 were enrolled. AF was defined as paroxysmal if AF terminated within 7 days of onset (spontaneously or with intervention). The exclusion criteria were as follows: poor quality of TEE image acquisition, reversible causes for AF, prior cardiac surgery, severe valvular disease, congenital heart disease and decompensated heart failure. The study design was approved by the Ethics Committee of Rehabilitation Hospital Cluj-Napoca and written informed consent prior to the procedure and participation in the study was obtained from each patient.

### 2.2. Transthoracic and Transesophageal Echocardiogram

All patients underwent both transthoracic echocardiography (TTE) and transesophageal echocardiography (TEE) within 24 h before RFCA. All the parameters were performed according to the American Society of Echocardiography guidelines. During TTE left ventricular end-systolic, and end-diastolic volumes (LVESV and LVEDV, respectively), and the left ventricular ejection fraction (LVEF) were measured. Left atrial diameter (LAD) was obtained on the parasternal long-axis view, while the left atrial volume (LAV) was calculated by the modified Simpson’s method. LAVI was obtained using body surface area correction. During TEE evaluation, LAA was visualized from the mid-oesophageal view at multiple omni-plane angles, between 45 and 100°. For pulsed-wave Doppler interrogation, the angle that provided the longest LAA dimension was used. The LAAeV was defined as a late diastolic positive outflow signal, and it was measured using a 4 mm sample volume positioned at the entry of the LAA orifice. In patients with paroxysmal AF at the time of TEE, LAAeV was described as the average value of five consecutive cardiac cycles (Figure 1). All cardiac ultrasound examinations were performed on a Philips Affiniti 50 (Philips Healthcare, Best, The Netherlands) with a 2–4 MHz microconvex transducer for ETT and a 7 MHz transducer for TEE. All echocardiographic measurements were conducted by the same two echocardiogram experts.

### 2.3. RFCA

Three-dimensional mapping of LA was performed with an Ensite NAVX Velocity (Saint-Jude Medical, Saint Paul, MN, USA) or a CARTO 3 (Biosense Webster, Diamond Bar, CA, USA) system. To create ablation lesions an open-irrigated 7-french 3.5 mm ablation catheter (Navistar Thermocol and Thermocool Smarttouch, Biosense Webster, Diamond Bar, CA, USA) or the FlexAbility irrigated ablation catheter (Saint-Jude Medical, Saint Paul, MN, USA) were used. Continuous radiofrequency ablation was performed by encircling the ipsilateral PVs until electrical isolation was achieved. The isolation of the electric field was considered to be complete in the absence of PV potential within each antrum, using a circular mapping catheter (LassoNav or PentaRay, Biosense Webster, Diamond Bar, CA, USA).

### 2.4. Follow-Up

All patients were scheduled for clinical visits to undergo 12-lead electrocardiography (ECG) and 24-h Holter recording at 3, 6, and 12 months after RFCA. Patients were also advised to present to our cardiology department if they experienced palpitations. AF recurrence was defined as any atrial tachyarrhythmia that lasted for more than 30 s captured by ECG or Holter monitoring. Transient episodes of arrhythmia recurrence during the first 3 months after ablation were counted as a blanking period.

### 2.5. Statistical Analysis

SPSS 21.0 statistical software (IBM, Chicago, IL, USA) was used to analyze the data. Continuous variables were checked for normality of distribution (Shapiro–Wilk test, kurtosis and skewness coefficients) and are presented as mean ± standard deviation, or median with interquartile range (25%, 75% percentiles). Categorical variables are described as frequency and percentage. Data were compared between the AF recurrence and non-recurrence groups using Mann–Whitney, Student’s t test or chi-square tests, when appropriate. Univariate and multivariate Cox proportional risk regression models were used to identify the risk factors associated with AF recurrence. ROC (receiver operating curve) analysis was used to calculate the best cut-off value of predictors of AF recurrence. Inter-observer reliability analysis was carried out using Cronbach’s alpha. A *p* value of <0.05 was considered statistically significant.

## 3. Results

A total of 81 consecutive patients with paroxysmal AF who underwent RFCA for the first time were enrolled, aged 55.3 ± 9 years, 48 (59.3%) men and 33 (40.7%) women. The median follow-up time was 12 (11;14) months. After the blanking period of 3 months, 24 patients (29.6%) had had AF recurrence (recurrence group) and 57 patients (70.4%) had maintained sinus rhythm (nonrecurrence group).

The clinical characteristics of patients with and without AF recurrence were compared in Table 1. There were no statistically significant differences in age, sex, body mass index (BMI), diabetes mellitus, hypertension, coronary heart disease, stroke, transient ischemic attack, duration of AF and medication between the two groups.

The LAD and LAVI were significantly greater in the recurrence than in the non-recurrence group (Table 2). The LAAeV was significantly (*p* < 0.05) lower for the patients with AF recurrence when compared to the patients who maintained a sinus rhythm (Table 2). However, no significant differences existed in the LVEDV, LVESV, and LVEF between the two groups. The Cronbach’s alpha for LAD and LAVI were 0.895 and 0.861, which shows good reliability. For LAAeV we calculated a Cronbach’s alpha of 0.904, which shows excellent reliability.

To investigate the associations between the various parameters and AF recurrence, a Cox proportional hazards regression analysis was performed. In the univariate analysis, high LAD and LAVI, as well as low LAAeV, were significant risk factors for AF recurrence (Table 3).

The variance inflation factors of LAD, LAVI, LAAeV were calculated and, because these factors were less than 5, no multiple collinearities existed (Table 4).

The variance inflation factors of LAD, LAVI, LAAeV were calculated and, because these factors were less than 5, no multiple collinearities existed (Table 4).

ROC curve analysis was used to evaluate the predictive efficiency of LAVI, LAAeV and LAD. The best cut-off value of the LAVI for the prediction of an AF recurrence was 40.5 mL/m^2^ with a sensitivity of 63%, specificity of 77% (AUC 0.776). An LAAeV cutoff value of 40.5 cm/s was selected for predicting rhythm outcome after catheter ablation with 89% sensitivity and 75% specificity (AUC 0.869). A LAD cutoff value over 41 mm was selected for predicting rhythm outcome after catheter ablation with 50% sensitivity and 84% specificity (AUC 0.670). The AUC for LAD was statistically significantly lower than the AUC for LAAeV (*p* = 0.001), but not than the AUC for LAVI (*p* = 0.146). The AUC for LAVI was not statistically significantly lower than the AUC for LAAeV (*p* = 0.184).

LAVI, LAAeV and LAD were introduced in multivariate analysis, after they were redefined according to their cut-off values (Table 5). LAAeV under 40.5 cm/s was the only independent predictor for recurrence after radiofrequency ablation of AF.

## 4. Discussion

Our study demonstrated that (1) LAD and LAVI were significantly elevated in patients with AF recurrence compared with those without recurrence, whereas LAAeV was significantly decreased in the recurrence than non-recurrence group; (2) the cut off points with the highest predictive efficacy for AF recurrence were LAAeV < 40.5 cm/s, LAVi < 40.5 mL/m^2^ and LAD > 41 mm; (3) only LAAeV was identified as independent predictor of AF recurrence after RFCA.

Even though RFCA is an effective therapeutic approach for patients with paroxysmal AF, recurrence of AF still remains problematic. In this study, the rate of AF recurrence following RFCA was 29.6%, which is consistent with previous research [10]. The recurrence rate after catheter ablation is higher for patients with persistent AF, and thus more risk factors have been established for AF recurrence [11,12]. We included only patients with paroxysmal AF in order to explore novel predictors of AF recurrence.

It is well known that LA enlargement is associated with AF recurrence after ablation. A meta-analysis of 22 studies has reported left atrium diameter to be a predictor of AF recurrence after ablation [5]. However, as LA dilation is asymmetric, the antero-posterior diameter is thought to underestimate LA size, whereas LA volume is more accurate in reflecting the size of the LA. Prior evidence showed that LAVI rather than LAD was related to AF recurrence following RFCA [13,14]. Although TTE can provide useful information for the detection of patients at risk for AF recurrence, LA volume is still determined using geometrical assumption, thus precisely estimating the LA structural dilation is challenging. Moreover, recent evidence shows that LA dysfunction is a more sensitive predictor for AF recurrence than LA size. Meanwhile, it is widely accepted that LAAeV measured by TEE can reflect not only LAA function but also LA function and severity of the LA remodeling. The LAA also presents contractile properties and a greater distensibility than the LA, that contributes to modulating LA pressure [15].

Kanda et al. used LAAeV as a surrogate factor of the LA function and were the first that demonstrated that a low LAAeV is associated with AF recurrence after catheter ablation [16]. They determined that the best cut-off value of the LAAeV for the prediction of AF recurrence was 28 cm/s (sensitivity 62%, specificity 69%). However, they included patients with persistent AF, probably with more severe LA remodeling compared to patients with paroxysmal AF, thus the LAAeV cut-off value they obtained is lower than in our study. Similar to our study were the results reported by He et al. that that also enrolled patients with paroxysmal AF and found that the best LAAeV cut off value in predicting AF recurrence was 39.2 cm/s (se 75, Sp 82) [17]. However, they showed that not only LAAeV but also LAVi and the left atrial appendage ejection fraction are significant factors to predict AF recurrence following radiofrequency ablation.

Xin-Xin et al. evaluated whether LAAeV and/or NT-proBNP levels are associated with AF recurrence after RFCA [18]. They showed that for patients with persistent AF, a low LAAeV combined with elevated plasma NT-proBNP concentrations can predict rhythm outcome following catheter ablation, while for patients with paroxysmal AF, a low LAAeV was the only independent predictor of CA efficacy.

Another study that included 193 patients who underwent successful electrical cardioversion found a similar cutoff for LAAeV of >40 cm/s measured by TEE that predicted long-term SR [19].

Patients with paroxysmal AF are usually in the early stages of LA remodeling. Chronic pressure overload causes LA enlargement, while impairment of LA function precedes LA dilatation. Therefore, functional parameters, such as LAAeV, may be more sensitive markers for AF recurrence after ablation than LA size, particularly in patients with paroxysmal AF. Our findings support this hypothesis, since LAVI and LAD were not associated with rhythm recurrence based on multivariate logistic regression and only LAAeV independently predicted AF recurrence.

Although the value of LAA in AF recurrence following radiofrequency ablation is of major importance, it has been scarcely explored. In practice, preprocedural observation of LAA by TEE is conventionally performed to detect thrombus formation. An LAAeV of less than 40 cm/s best predicts thromboembolic risk, while a value of more than 40 cm/s is correlated with successful AF cardioversion and sinus rhythm maintenance [20,21,22]. However, there is no clear cutoff that would contraindicate the ablation procedure. The ablation is only contraindicated if an LAA thrombus is found because the procedure poses a risk of thromboembolic stroke due to thrombus mobilization caused by catheter manipulation. If the LAAeV is less than 40 cm/s and additionally spontaneous contrast is present in the LAA, a CT or intracardiac echocardiogram should be performed before ablation to rule out an LAA thrombus. Therefore, when LAAeV is less than the cut-off value of 40.5, it should not only raise the concern of a thrombus formation but also it might identify patients at risk to develop AF recurrence after RFCA. Moreover, if further studies on larger cohorts find a similar high AF recurrence rate based on low LAAeV, physicians may consider not performing RFCA. Identifying new risk predictors of AF recurrence is critical as it may contribute to a better selection of patients to undergo RFCA and to improve the success rate of catheter ablation.

The current study presented several limitations. Although we closely monitored patients using periodic ECG and Holter monitoring, asymptomatic episodes of AF may have been missed, which may have resulted in underestimation of the AF recurrence rate. We included a small cohort of patients, and thus our preliminary findings should be validated by future research including a larger number of patients.

## 5. Conclusions

In summary the left atrial diameter and the left atrial volume were significantly increased, while the left atrial appendage emptying flow velocity was significantly decreased in recurrence than in non-recurrence after radiofrequency ablation. A low left atrial appendage emptying flow velocity was the only independent predictor of AF recurrence within 1 year after radiofrequency ablation.

## Figures and Tables

**Figure 1 diagnostics-11-01465-f001:**
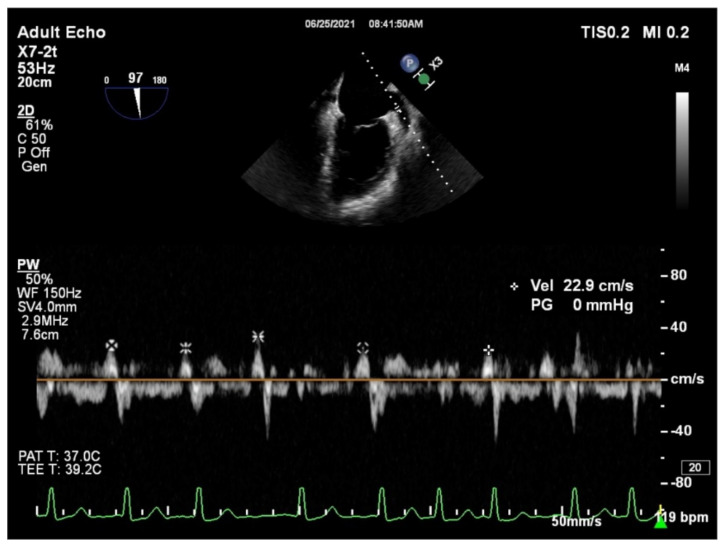
Example of measuring the left atrial appendage emptying velocity (LAAeV). The LAAeV was measured by pulsed-wave Doppler interrogation at the entry of the left atrial appendage at 84° (22.9 cm/s).

**Table 1 diagnostics-11-01465-t001:** Clinical characteristics after RFCA of AF.

Baseline Characteristics		Recurrence N = 24	Non-Recurrence N = 57	*p*
Age (years)		55.9 ± 9.8	54.6 ± 8.4	0.5
Male, *n* (%)		12 (50)	31 (54.4)	0.37
BMI (kg/m^2^)		26.6 ± 2.5	25.9 ± 2.8	0.27
Coronary artery disease		5 (20.8)	4 (7)	0.07
Hypertension, *n* (%)		11 (45)	22 (38.6)	0.5
Diabetes mellitus, *n* (%)		3 (12.5)	4 (7)	0.42
Stroke or TIA, *n* (%)		3 (12.5)	2 (3.5)	0.12
Hyperlipidaemia, *n* (%)		7 (29.2)	17 (29.8)	0.49
AF history (months)		34 (10, 55)	36 (11, 45)	0.38
**Antiarrhythmic Drug**	Propafenone, *n* (%)	5 (20.8)	12 (21.1)	**0.127**
	Amiodarone, *n* (%)	5 (20.8)	22 (38.6)	
	Flecainide, *n* (%)	11 (45.8)	12 (21.1)	
Betablocker, *n* (%)		12 (50)	20 (35.1)	0.315
RAS inhibitors, *n* (%)		11 (45.8)	23 (40.4)	0.834
Statin, *n* (%)		11 (45.8)	25 (45.6)	1

**Table 2 diagnostics-11-01465-t002:** Ultrasound parameters.

Ultrasonic Parameters	Recurrence N = 24	Non-Recurrence N = 57	*p*
LAD (mm)	39.8 ± 5.7	37.1 ± 4.6	0.03
LAVI (mL/m^2^)	43 (37; 51)	36 (32; 40)	<0.001
LVEDV (mL)	92 (82; 101.5)	94.5 (83.5; 110.75)	0.16
LVESV (mL)	33 (27.5; 43.5)	35 (29.5; 58.5)	0.13
LVEF (%)	61 (55; 64.7)	60 (55; 63)	0.497
LAAeV (cm/s)	37.9 ± 8.5	49.4 ± 6.8	0.001

**Table 3 diagnostics-11-01465-t003:** Univariate analysis for AF recurrence.

	HR (95% CI)	*p*
Age	1.021 (0.974–1.072)	0.386
Gender	1.565 (0.703–3.484)	0.273
Coronary artery disease	2.626 (0.979–7.042)	0.055
Hypertension	1.263 (0.566–2.820)	0.568
Diabetes mellitus	1.655 (0.493–5.557)	0.415
Stroke	3.944 (0.872–17.833)	0.075
Hyperlipidaemia	1.238 (0.449–3.415)	0.680
Propafenone	1.346 (0.322–5.634)	0.684
Flecainide	0.787 (0.188–3.295)	0.743
Amiodarone	2.290 (0.638–8.212)	0.204
Betablocker	1.638 (0.736–3.647)	0.227
RAS inhibitors	1.209 (0.541–2.698)	0.644
Statin	1.015 (0.455–2.266)	0.971
LAD	1.112 (1.025–1.207)	0.011
LAVi	1.126 (1.072–1.183)	<0.001
LVEF	1.031 (0.948–1.122)	0.475
LVEDV	1.022 (0.993–1.051)	0.142
LVESV	1.027 (0.998–1.057)	0.068
LAAeV	0.856 (0.807–0.908)	<0.001

**Table 4 diagnostics-11-01465-t004:** Collinearity analysis.

	LAD	LAVI	LAAeV
**VIF**	1.842	1.693	1.671

**Table 5 diagnostics-11-01465-t005:** Multivariate analysis for AF recurrence.

	B	*p*	HR	95.0% CI for HR	
				Min	Max
Age	−0.025	0.375	0.976	0.924	1.030
Gender (female)	0.852	0.071	2.345	0.930	5.913
LAD > 41 mm	0.731	0.165	2.077	0.741	5.820
LAVI > 40.5 mL/m^2^	0.358	0.467	1.430	0.545	3.755
LAAeV < 40.5 cm/s	2.150	<0.001	8.588	3.125	23.599

Abbreviations: B, unstandardized regression coefficient; P, coefficient of statistical significance; HR, hazard ratio; CI, confidence interval.

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
