# Peer review of "Value of Left Atrial Appendage Function Measured by Transesophageal Echocardiography for Prediction of Atrial Fibrillation Recurrence after Radiofrequency Catheter Ablation"

_diagnostics, 2021, doi:10.3390/diagnostics11081465_

Round 1

Reviewer 1 Report

Authors mostly addressed the reviewer’s comments. The following points should be revised further.

Lines 31, Abstract, “LAVI >40.ml”  may be “LAVI >40.5ml. This should be revised correctly.

Lines 91, Figure legends, “52 cm/s” may be “22.9 cm/s”. This should be revised correctly.

Table 5, No abbreviation was presented with the footnote.

Author Response

Comment no 1. "Lines 31, Abstract, “LAVI >40.ml”  may be “LAVI >40.5ml. This should be revised correctly."

Response to comment no 1: Thank you for your observation. We corrected the LAVI cutoff. 

Comment no 2: "Lines 91, Figure legends, “52 cm/s” may be “22.9 cm/s”. This should be revised correctly."

Response to comment no 2: Thank you. We have revised the figure legend correctly.

Comment no 3: "Table 5, No abbreviation was presented with the footnote."

Response to comment no 3: Thank you. We have added the abbreviations. 

Reviewer 2 Report

The authors have significantly improved the quality of their work and I consider that the manuscript can now be reconsidered for publication in its current version. 

Author Response

Thank you for all your valuable input. 

This manuscript is a resubmission of an earlier submission. The following is a list of the peer review reports and author responses from that submission.

Round 1

Reviewer 1 Report

I have read with interest the article of Istratoaie and coworkers, regarding the value of left atrial appendage function in predicting atrial fibrillation recurrence after radiofrequency catheter ablation. It is known that radiofrequency catheter ablation is an effective and potentially curative treatment in patients with paroxysmal atrial fibrillation, but, unfortunately, the success rate ranges between 38% and 62%. This study aimed to assess the value of left atrial appendage emptying flow velocity in predicting AF recurrence following radiofrequency catheter ablation in paroxysmal AF patients.

I consider that the article may be accepted for publication after some minor revisions.

  1. I suggest emphasizing much better the novelty of their study, considering that their results are similar to those reported by He et al.
  2. I would like to ask the authors if they also took into account the results obtained from the radiofrequency catheter ablation procedure (including periprocedural complications) on the risk of atrial fibrillation recurrence.

Author Response

Response to Reviewer 1 Comments

Point 1:  I suggest emphasizing much better the novelty of their study, considering that their results are similar to those reported by He et al.

Response 1: Thank you for your observation. Upon closer examination of the methods, we modified the approach. We calculated the cut-off value also for LAD. The multivariate analysis was remade with LAD, LAVI and LAAeV introduced as nominal variables according to their cut-off values. We concluded that LAAeV under 40.5 cm/s was the only independent predictor for recurrence after radiofrequency ablation of AF. In contrast to other studies, although LAD and LAVI were significantly elevated in patients with AF recurrence, these parameters were not associated with rhythm recurrence based on multivariate logistic regression.  

We speculated and added to the text: “Given that we included only patients with paroxysmal AF, it could be argued that the LA was not that dilated for LAVI and LAD to be independent predictors for AF recurrence. In our study LAAeV  was a more sensitive predictor for AF recurrence . Thus greater focus should be placed on evaluation of LAA function as a predictive factor for AF recurrence after RFCA, since a low LAAeV value that reflects LA function might precede the onset of LA dilation.”

Moreover we discussed about whether RFCA indication changes when LAAeV is less than the cut-off value of 40.5:

“ LAAeV under 40 cm/s best predicts thromboembolic risk, while a value greater than 40cm/s is correlated with successful cardioversion of AF and maintenance of sinus rhythm[20,21,22]. However there is no definite cut-off that would contraindicate the ablation procedure. Only if a LAA  thrombus is detected, the ablation is contraindicated, since the procedure carries a risk of thromboembolic stroke due to mobilisation of the thrombus secondary to catheter manipulation. On the other hand, if the LAAeV is <40cm/s and additionally spontaneous contrast is present in the LAA, it is recommended to perform a CT or intracardiac echocardiography to rule out a LAA thrombus before ablation. Therefore when LAAeV is less than the cut-off value of 40.5, it should not only raise the concern of a thrombus formation but also it might identify patients at risk to develop AF recurrence after RFCA”

Point 2: I would like to ask the authors if they also took into account the results obtained from the radiofrequency catheter ablation procedure (including periprocedural complications) on the risk of atrial fibrillation recurrence.

Response 2: Thank you for you question. In our small cohort of patients with paroxysmal AF we had no periprocedural complications. The time of procedure also was similar, thus we have not taken into consideration RFCA procedure characteristics. If in our study were also patients with persistent AF, we would have analzyed whether the procedure characteristics have an impact on AF recurrence. 

Reviewer 2 Report

For the present study, Istratoaie et al. investigated the interactions between the left atrial appendage function and AF recurrence in paroxysmal AF patients after RFCA. The authors found that a lower LAAeV was significantly associated with a higher AF recurrence.

This study is potentially interesting, but there were critical issues to be addressed regarding study design, methodologies, analysis, and interpretation of the data. See the specific comments as summarized below.

  1. Novelty should be improved. It is well established that LAAeV is associated with LA remodeling, which predicts the Af recurrence after ablation shown in the priors studies of ref 16-22 as the authors stated in the Discussion.

  1. The numbers of the enrolled subjects were very small. Authors need to consider this study as preliminary.

  1. It is unclear the LAAeV was measured at sinus rhythm or Af rhythm. This is very important.

  1. Did the authors define that LAAeV was defined of the velocities of inflow and/or outflow of LAA?

  1. In Figure 1, the measurement point of LAAeV was not precisely measured. The LAAeV level is unlikely 52 cm/s. No ECG was displayed. Five consecutive cardiac cycles were not averaged.

  1. Medications were not considered for the analysis.

  1. When LAAeV was less than the cut-off value of 40.5, how RFCA indication could be considered? No discussion was demonstrated for this.

  1. Lines 183-185; Our study results support that LAAeV was superior to LAD and LAVI as a predictor of AF recurrence, while the area under the ROC curve (AUC) for LAAeV was greater than that for LAVI (AUC 185 0.869 vs. 0.776). This is overstated because the authors did not show statistical comparisons between them.

Author Response

Before addressing each of the comments below, the authors would like to thank the reviewer for his/her time and valuable comments. They raise important issues and their inputs are very helpful for improving the manuscript. We agree with almost all their comments and we have revised our manuscript accordingly.

The modifications we have made in the revised manuscript are marked in track changes. We added new text, images and statistical tests.

Please, find below the referees’ comments repeated and our responses in red color inserted in after each comment.

COMMENTS TO AUTHOR:

Reviewer #2:

  1. Novelty should be improved. It is well established that LAAeV is associated with LA remodeling, which predicts the Af recurrence after ablation shown in the priors studies of ref 16-22 as the authors stated in the Discussion.

Response to Comment No 1:

Thank you for your suggestion. Upon closer examination of the methods, we modified the approach and we concluded that LAAeV under 40.5 cm/s was the only independent predictor for recurrence after radiofrequency ablation of AF. In contrast to other studies, although LAD and LAVI were significantly elevated in patients with AF recurrence, these parameters were not associated with rhythm recurrence based on multivariate logistic regression. We speculated and added to the text: “Given that we included only patients with paroxysmal AF, it could be argued that the LA was not sufficiently dilated for LAVI and LAD to be independent predictors of AF recurrence. In our study LAAeV was a more sensitive predictor for AF recurrence. Since a low LAAeV value that reflects LA function may occur before LA dilation onset, greater emphasis should be placed on assessment of LAA function as a predictor of AF recurrence. “

  1. The numbers of the enrolled subjects were very small. Authors need to consider this study as preliminary.

Response to Comment No 2:

Thank you for your observation. We added as limitation that our data is preliminary and further research is needed.

  1. It is unclear the LAAeV was measured at sinus rhythm or Af rhythm. This is very important.

Response to Comment No 3:

Thank you for this valid observation. We cleared that only when the patient was in AF during TEE, LAAeV was measured as the average value of five consecutive cardiac cycles.

  1. Did the authors define that LAAeV was defined of the velocities of inflow and/or outflow of LAA?

Response to Comment No 4:

Thank you for this comment. We defined LAAeV as a late diastolic positive outflow signal.

  1. In Figure 1, the measurement point of LAAeV was not precisely measured. The LAAeV level is unlikely 52 cm/s. No ECG was displayed. Five consecutive cardiac cycles were not averaged.

Response to Comment No 5:

Thank you for your suggestion. We replaced the figure.

  1. Medications were not considered for the analysis.

Response to Comment no 6:

Thank you for your observation. We have not considered relevant medication in this study, since our patients were in paroxysmal AF and not all of them followed antiarrhythmic therapy. Also since there were no difference in comorbidities between the recurrence and non recurrence group, we did not include medication for comorbidities. However, we are open to include medication if you consider this relevant.

  1. When LAAeV was less than the cut-off value of 40.5, how RFCA indication could be considered? No discussion was demonstrated for this.

Response to Comment no 7:

 Thank you for your suggestion. We added the following:

“LAAeV under 40 cm/s best predicts thromboembolic risk, while a value greater than 40cm/s is correlated with successful cardioversion of AF and maintenance of sinus rhythm[20,21,22]. However there is no definite cut-off that would contraindicate the ablation procedure. Only if a LAA  thrombus is detected, the ablation is contraindicated, since the procedure carries a risk of thromboembolic stroke due to mobilisation of the thrombus secondary to catheter manipulation. On the other hand, if the LAAeV is <40cm/s and additionally spontaneous contrast is present in the LAA, it is recommended to perform a CT or intracardiac echocardiography to rule out a LAA thrombus before ablation. Therefore when LAAeV is less than the cut-off value of 40.5, it should not only raise the concern of a thrombus formation but also it might identify patients at risk to develop AF recurrence after RFCA”

  1. Lines 183-185; Our study results support that LAAeV was superior to LAD and LAVI as a predictor of AF recurrence, while the area under the ROC curve (AUC) for LAAeV was greater than that for LAVI (AUC 185 0.869 vs. 0.776). This is overstated because the authors did not show statistical comparisons between them.

Response to Comment no 8:

Thank you for the observation. Upon closer examination of the methods, we modified the approach. We calculated the cut-off value also for LAD. The multivariate analysis was remade with LAD, LAVI and LAAeV introduced as nominal variables according to their cut-off values. We added to the text the following :

A LAD cutoff value over 41 mm was selected for predicting rhythm outcome after catheter ablation with 50% sensitivity and 84% specificity (AUC 0. 670).

LAVI, LAAeV and LAD were introduced in multivariate analysis, after they were redefined according to their cut-off values (Table 5). LAAeV under 40.5 cm/s was the only independent predictor for recurrence after radiofrequency ablation of AF.

Table 5. Multivariate analysis for AF recurrence

B

P

HR

95.0% CI for HR

Min

MAX

LAD >41 mm

0.315

0.507

1.370

0.541

3.473

LAVI >40.5 ml/m2

0.484

0.319

1.623

0.626

4.206

LAAeV <40.5 cm/s

2.103

<0.001

8.194

2.980

22.530

Round 2

Reviewer 2 Report

The authors have replied to some of the reviewer’s concerns, but the remaining issues are seriously raised and should be addressed further. Unless these are addressed adequately, authors cannot reach the conclusion they made in this manuscript.

  1. Comment #1; ”it could be argued that the LA was not sufficiently dilated for LAVI”. Please clarify the intended meaning. This description is unclear. English languages should be intensively improved throughout the manuscript.
  2. Comment #6; Medications including antiarrhythmic therapy and RAS inhibitors, beta-blockers, statins may have an impact on AF recurrence. The authors should have shown the ratios/contents of medical therapies and should have considered them as important factors for AF recurrence. If no medications were given to all patients, these facts should have been shown.
  3. Comment #7; Authors mentioned that “When LAAeV was less than the cut-off value of 40.5, it might identify patients at risk to develop AF recurrence after RFCA”. This is not sufficiently discussed. Given the recurrence rate is very high based on LAAeV with HR of 8.1, authors may consider not performing RFCA or consider specific medications to AF in clinical practice.
  4. Comment #8; Multivariate analysis for AF recurrence was not applied scientifically. For example, age and sex were not considered as covariates for the analysis.
  5. Comment #8; Authors should have performed comparing correlated ROC curves in terms of the area under the curve (AUC) for AF recurrence.
  6. The abbreviations were not shown (eg, B)

Author Response

Before addressing each of the comments below, the authors would like to thank the reviewer for his/her time and valuable comments. We agree with all their comments and we have revised our manuscript accordingly.

The modifications we have made in the revised manuscript are marked in track changes. We added new text and statistical tests.
Please, find below the referees’ comments repeated and our responses in red color inserted in after each comment.

COMMENTS TO AUTHOR:
Reviewer #2:
1. ”it could be argued that the LA was not sufficiently dilated for LAVI”. Please clarify the intended meaning. This description is unclear. English languages should be intensively improved throughout the manuscript.

Response to Comment No 1:
Thank you for your valid observation. We added the following:
“Patients with paroxysmal AF are usually in the early stages of LA
remodeling. Chronic pressure overload causes LA enlargement, while
impairment of LA function precedes LA dilatation. Therefore functional
parameters, such as LAAeV, may be more sensitive markers for AF
recurrence after ablation than LA size, particularly in patients with
paroxysmal AF. Our findings supports this hypothesis, since LAVI and
LAD were not associated with rhythm recurrence based on multivariate
logistic regression and only LAAeV independently predicted AF
recurrence.”
We hope we have improved English language throughout the
manuscript.

  1. Medications including antiarrhythmic therapy and RAS inhibitors, beta-blockers, statins may have an impact on AF recurrence. The authors should have shown the ratios/contents of medical therapies and should have considered them as important factors for AF recurrence. If no medications were
    given to all patients, these facts should have been shown.

Response to Comment No 2:
Thank you for your suggestions. We added medication including antiarrhythmic drugs (Class I and III, beta-blockers, RAS inhibitors and statins in table 1 and table 3 analysis.

  1. Comment #7; Authors mentioned that “When LAAeV was less than the cut-off value of 40.5, it might identify patients at risk to develop AF recurrence after RFCA”. This is not sufficiently discussed. Given the recurrence rate is very high based on LAAeV with HR of 8.1, authors may consider not performing RFCA or consider specific medications to AF in clinical practice.

Response to Comment No 3:
Thank you for this suggestion. We have proposed exactly your suggestion if further studies would validate our results.

  1. Comment #8; Multivariate analysis for AF recurrence was not applied scientifically. For example, age and sex were not considered as covariates for the analysis.

Response to Comment No 4:
Thank you for this comment. We included age and sex as covariates for the multivariate analysis (Table 5)

  1. Comment #8; Authors should have performed comparing correlated ROC curves in terms of the area under the curve (AUC) for AF recurrence.

Response to Comment No 5:
Thank you for your suggestion. We compared the ROC curves and added the following:
“A LAD cutoff value over 41 mm was selected for predicting
rhythm outcome after catheter ablation with 50% sensitivity and
84% specificity (AUC 0. 670). The AUC for LAD was statistically
significant lower than the AUC for LAAeV (p=0.001), but not than
the AUC for LAVI (p=0.146). The AUC for LAVI was not
statistically significant lower than the AUC for LAAeV (p=0.184).

  1. The abbreviations were not shown (eg, B)

Response to Comment no 6:
Thank you for your observation. We have shosn the abbreviations (BMI, LA, LAA).